# The Epidemiological and Economic Impact of COVID-19 in Kazakhstan: An Agent-Based Modeling

**DOI:** 10.3390/healthcare11222968

**Published:** 2023-11-16

**Authors:** Berik Koichubekov, Aliya Takuadina, Ilya Korshukov, Marina Sorokina, Anar Turmukhambetova

**Affiliations:** 1Department of Informatics and Biostatistics, Karaganda Medical University, Gogol St. 40, Karaganda 100008, Kazakhstan; takuadina@qmu.kz (A.T.); korshukov@qmu.kz (I.K.); m.sorokina@qmu.kz (M.S.); 2Institute of Life Sciences, Karaganda Medical University, Gogol St. 40, Karaganda 100008, Kazakhstan; turmuhambetova@qmu.kz

**Keywords:** epidemiology, COVID-19, agent-based model, forecasting

## Abstract

Background: Our study aimed to assess how effective the preventative measures taken by the state authorities during the pandemic were in terms of public health protection and the rational use of material and human resources. Materials and Methods: We utilized a stochastic agent-based model for COVID-19’s spread combined with the WHO-recommended COVID-ESFT version 2.0 tool for material and labor cost estimation. Results: Our long-term forecasts (up to 50 days) showed satisfactory results with a steady trend in the total cases. However, the short-term forecasts (up to 10 days) were more accurate during periods of relative stability interrupted by sudden outbreaks. The simulations indicated that the infection’s spread was highest within families, with most COVID-19 cases occurring in the 26–59 age group. Government interventions resulted in 3.2 times fewer cases in Karaganda than predicted under a “no intervention” scenario, yielding an estimated economic benefit of 40%. Conclusion: The combined tool we propose can accurately forecast the progression of the infection, enabling health organizations to allocate specialists and material resources in a timely manner.

## 1. Introduction

The COVID-19 pandemic has been a colossal test of strength for healthcare systems worldwide, including in Kazakhstan. The Republic implemented unprecedented measures to support the healthcare system, which significantly reduced the spread of infection, morbidity, and mortality. Recent years have shown that understanding the dynamics by which the disease spreads and proper decision-making is vital for combating and containing the disease. For more than a hundred years, mathematicians and later epidemiologists have invested their efforts in designing models that could predict epidemics statistically. In addition to predicting the future state of a pandemic and the number of infections involved, policymakers need a model to determine the necessary and optimal decisions.

Classic models describe the dynamics of transmission using systems of differential equations. A significant advantage of models based on differential equations is their potential for analysis. Epidemiological compartmental models are a very general modeling technique. The population is assigned to compartments with labels—for example, S, I, or R (susceptible, infectious, or recovered). People can move between compartments. The order of the compartments usually reflects the flow patterns between them; for example, SEIS means susceptible, exposed, infectious, and then susceptible again. There are many modifications to this model. Some examples include the classic Susceptible–Infect–Susceptible (SIS) epidemic model established by Kermack and McKendrick, the Susceptible–Infect–Recovered (SIR) epidemic model proposed by Bailey, the Susceptible–Infect–Vaccination–Susceptible (SIVS) epidemic system introduced by Arion et al., and the stochastic Susceptible–Infect–Quarantine–Susceptible (SIQS) epidemic model studied by Zhang et al. [1]. However, they cannot simultaneously and effectively model the spread of infection in space and time, the specifics of individual contact, account for all the details of individual behavior, etc. In this regard, computer simulation models are currently favored.

Deterministic models based on differential equations have been replaced by models based on cellular automata, as well as models that take into account the stochastic nature of disease transmission [2,3]. The main disadvantage of these models is that they do not consider the mobility of people and their interaction in space, which is an important factor in highly contagious diseases. Therefore, other methods have been developed.

Recently, researchers have become increasingly attracted to artificial intelligence to solve complex problems in several fields, including medicine. Intelligent systems based on machine learning (ML) allow for the combination of information and observations from different sources in order to identify the data related to the object of research, search for patterns, classify the data, and make predictions based on the obtained model. ML algorithms can read and modify their structure based on the observed data set, adapting by optimizing the target function. ML is widely used with COVID-19, including in prevalence analysis, diagnosis, prognosis, public health, clinical decision-making, preventative measures, vaccine development, and surveillance [4]. Presently, machine learning provides extremely important tools for intelligent geo- and environmental data analysis, processing, and visualization [5].

Agent-based models (ABMs) are similar to cellular automata-based models, but they leverage the extra tracking of social interactions’ effects on individual entities. Agent-based models are complex systems of autonomous agents with their own goals, behaviors, and interactions with each other and the environment. As in the real world, global behavior and trends emerge from the behavior and interactions of individual agents [6,7].

To better understand how ABM can be used in pandemics, Lorig et al. [8] conducted a systematic review to identify agent-based models of COVID-19 transmission. They analyzed 126 publications on the subject. The models were developed in the early stages of the spread of the infection. There are simplified behavior models and more advanced dynamic or adaptive behavior models. The main purpose of the models is to predict the dynamics of COVID-19 over time and develop emergency interventions for crises. It has been shown that the prediction period varies greatly between the models. It varies from a few days to 2 years. However, most of the models make predictions between three months and one year. This paper discusses the problematic issues and, in particular, the extent to which the models reflect the real world and whether they are suitable for making appropriate decisions.

Many researchers use the methodology of Covasim (a COVID-19 agent-based simulator) in their work [9]. This tool includes extensive information on population demographics, transmission sites, and incidence data using age and virus strain. Covasim also analyzes the impact of drug and non-drug interventions.

Agent-based modeling is a convenient tool for analyzing the life activities of medium-sized communities. The creation of different virtual populations with the accurate representation of buildings, public places, and transport allows for a thorough analysis of the epidemiological situation and the development of targeted measures to prevent the spread of infection [10].

One advantage of agent-based modeling is the ease of modeling various scenarios [11,12,13]. For example, Braun et al. [14] and Kai et al. [15] simulated the universal use of masks combined with social distancing in their models. Based on the experience of applying this approach, the testing policy [16], strategy for reopening public places [17], effectiveness of treatment methods, and development of a vaccination strategy were evaluated [18].

Some articles in the literature have used agent-based models to simulate the economic and epidemiologic impacts of COVID-19. For example, a study was conducted on the epidemiological and economic impact of COVID-19 in the United States. The results show that the trade-offs between economic losses, lives saved, and infections averted are non-linear with respect to social-distancing adherence and lockdown duration. The most impacted industries were not labor-intensive industries, such as the agricultural and construction sectors, but those in which jobs are highly valued, such as professional services [19].

Inoue and Todo [20] calculated that a one-month lockdown in Tokyo would result in a total production loss of 5.3% of Japan’s annual gross domestic product (GDP). Dignum et al. [21] proposed a tool to analyze the pandemic’s health, social, and economic impacts when the government implements a range of interventions, such as closing schools, requiring employees to work from home, and providing subsidies to the public. Silva PCL et al. [22] found that the best solution would be the “use of face masks and 50% social isolation” in the event of a full lockdown. This scenario is optimal from the perspective of minimizing the number of deaths and saving business, government, and people’s revenues.

Zhang et al. [23] assessed the need for medical resources based on different scenarios involving COVID-19 spread and intervention. The authors made recommendations based on investments in healthcare and the allocation of resources. Moynihan R et al. highlighted the extent and nature of changes in healthcare during the COVID-19 pandemic in a systematic review [24].

Other authors have developed COVID-19 strategies that may be most beneficial in resource-limited settings [25].

Most researchers agree that ABM is used as an alternative to classical mathematical tools. This approach allows us to assess how patterns of disease transmission vary across different population characteristics and influence the overall dynamics of the epidemic. ABM has significant potential and flexibility in modeling the interactions between individuals as well as their movements, given that the COVID-19 outbreak was highly dependent on these two parameters. Importantly, it can be integrated with geospatial information science (GIS), which therefore enables spatial modeling of the phenomenon. On the other hand, ABMs are models in which various strategies can be considered and analyzed, and decision-making can be optimized. They are powerful tools for simulating operational activities and strategic management [26].

Despite their widespread use, ABMs are subject to several limitations. In particular, they are very sensitive to initial conditions and small changes in the rules of interaction. This is especially true for models in which agents are humans. Such systems may include agents with potentially irrational behavior, subjective opinions, and complex psychology. These factors are difficult to measure and calibrate. Attempting to capture the full variety of interpersonal interactions makes the model more complex to implement, requires more computer resources, and makes the results more difficult to interpret [27].

One of the goals of modeling is to predict the spread of COVID-19 over time. However, we faced several problems that did not allow reliable predictions of morbidity. One review devoted to analyzing various models showed that out of 59 predictions, the predicted values were higher in 38% of cases than the real ones and lower than the observed values in 62% of cases. Differences in accuracy were not found between different categories of models nor within each category [8].

This discrepancy may be due to the validity of the proposed models. Heath et al. [28] showed that 65% of the agent-based models presented in the articles did not pass rigorous testing. Another review [29] based on an analysis of 126 agent-based models showed that most articles (about 87%) did not specify how their models or results were tested. Models are often difficult to test due to the unavailability or incompleteness of real data.

Another reason is the high variability in the daily number of COVID-19 cases. The transmission of an infectious disease in a population is a complex phenomenon. Behavior is mainly influenced by interactions between people and not just individual characteristics [30]. Interactions can be viewed as social processes [31], such as contact between people and place effects [32]. Interaction complexity also includes effects over time [33], where the past, present, and future affect the decision context; for example, persons already found to be infected are isolated and no longer represent a source of infection. In addition, other factors influence the rate, such as the seasonality of pneumonia, mobility, transmission frequency, weather-dependent use of masks, social behavior, stress, public health measures, etc. During the pandemic, countries implemented different preventative measures, and the level of compliance with these measures differed. Therefore, COVID-19 transmission has local characteristics.

Therefore, determining the transmission rate is an important step in creating a model for disease transmission. Some authors rely on the contact probability matrix [34], whereas others determine the physical proximity of agents using geolocation [15]. Many COVID-19 models have estimated time-varying rates of transmission based on case counts or hospitalization data. In some cases, models have inferred changes in transmission rates using estimates of the pathogen’s instantaneous reproduction number [35,36,37].

In this regard, several authors have considered stochastic models of the spread of COVID-19 [38,39,40,41,42,43,44,45,46]. Various scenarios of preventative measures were considered in stochastic ABMs, including the environment’s influence on the spread of COVID-19 [18,46]. One of the advantages of the stochastic model is that it considers various COVID-19 properties, such as the unpredictability of sudden outbreaks of disease, long periods of asymptomatic courses, and sudden declines followed by bursts [47].

Based on the above, we can conclude that insufficient accuracy of COVID-19 forecasting is associated with the following problems:A significant part of the models lacks validity checks due to the incompleteness of real data.Most of the models were created in the early stages of the pandemic. When predicting, the researchers proceeded from the assumption of a further increase in the number of new cases. As further developments have shown, there may be periods of relative stability that are accompanied by sudden outbreaks, the nature of which has not yet been explained.Some models did not take into account the high variability in the daily number of COVID-19 cases associated with the fact that the transmission of an infectious disease in a population is a complex phenomenon.

These shortcomings were also inherent in the model proposed by Kazakhstan’s scientists earlier. As a result, as time has shown, the forecast data significantly differed from reality.

The aim of our study was to assess how effective the preventative measures taken by the state authorities were during the pandemic in terms of public health protection and rational use of material and human resources.

This paper is structured as follows:

We created a stochastic agent-based model of the spread of COVID-19 for a large city in Kazakhstan, in which we tried to take into account the above-mentioned problems (Section 2.1 and Section 2.2).

We used the OptQuest optimizer, included in AnyLogic software, to validate a stochastic ABM for the spread of COVID-19. Then, we considered the model’s forecasting accuracy (Section 3.1 and Section 3.2).

To assess the social impact of the pandemic, we identified the population groups most susceptible to infection in five different hypothetical scenarios as well as in real-life situations. Public places with the greatest risk of transmission were also identified. In this way, we assessed the effectiveness of the measures taken by the regulatory authorities.

We combined the created model with the material and labor cost estimation tool recommended by WHO. Based on this combination, we assessed how preventative measures implemented by the regulators affect material and human resources compared to six hypothetical scenarios: no intervention, school clause, mask-wearing, vaccination, and combined measures (Section 3.3 and Section 3.4).

## 2. Materials and Methods

### 2.1. Study Area

We considered a stochastic agent model of the spread of COVID-19 for the fourth largest settlement in Kazakhstan, Karaganda. It is the Republic’s major industrial center. The number of social infrastructure facilities in the city is given in Table 1.

### 2.2. ABM Model Implementation

The model consists of two parts (state charts). The first part (social network) reflects the daily social activity of the city population, consisting of staying in the family, at work, in educational institutions, and other places where person-to-person contact can occur. In this case, there is the possibility of transmission from a sick person to a healthy person. Further spread of the disease and the behavior of the sick person occur according to the second part of the model (disease transmission), which reflects the process of diagnosis and treatment.

The model was created in AnyLogic University Researcher 8.8.1 software. Chicago, United States. A demo version of the model at a 1:10 scale is available at https://cloud.anylogic.com/model/ba844cef-a41f-4032-8e1f-517e243b8e5f?mode=SETTINGS (accessed on 15 August 2023). It includes two state charts: social network and disease transmission.

#### 2.2.1. Social Network

We divided the entire population into five groups: children attending preschool, children attending other school institutions, university students, working adults, and non-working adults. Each group has its own pattern of behavior. Families are randomly filled with individuals from these categories. The one-person, two-person, three-person, four-person, and five-person families correspond to the official data on the stat.gov.kz website.

Given that modeling social contact is a non-trivial task, we have simplified contacts at both the individual level and at the level of families and public places. This allowed us to model different types of social contacts throughout the day (Figure 1).

By social contact, we mean close contact over time with a certain number of people. This could be relatives or people meeting at school, on transport, in a shop, or at work. Brief contact between people in other public places was also considered. We assumed that schools, preschool institutions, shops, and clinics were within walking distance, whereas university students used public transportation. Preschoolers, pupils, and university students were in contact with their groups and classes. Working contact was observed among persons 26–60 years old. According to the classifications existing in Kazakhstan, we distinguished between small, medium, and large enterprises. Their number corresponds to statistics. We conducted the filling of enterprises randomly. We assumed that for non-working persons, the public places they visit are mainly outpatient clinics, convenience stores, and shopping centers where transmission of infection from carriers to healthy persons can occur. Other infection sources include their family, public transport, work, and educational institutions.

#### 2.2.2. Disease Transmission

The spread of infection follows the well-known SEIR model and depends on the number of carrier contacts and the probability of infection.

At the start of the pandemic, agents were in a “susceptible” state, except for a few initial infections (Figure 2).

Agents move or stop according to their own trajectory. Once susceptible agents meet infectious agents and become infected, the state of susceptible agents changes to “exposed.” After a given “latent period,” those individuals become “infectious” and can infect other agents. COVID-19 symptoms appear during the incubation period after exposure. After diagnosis, some of the sick remain on outpatient treatment, are isolated at home, and recover soon. The remainder are sent to the hospital, where they might be moved to intensive care with artificial lung ventilation if conditions worsen. The outcomes for all patients are recovery or death. Once recovered, the agents can no longer be infectious and cannot be reinfected. We assumed this trend early in the pandemic when the risk of reinfection was low [48].

#### 2.2.3. Model Parameters

Table 2 presents the socioeconomic and epidemiological parameters of the model.

The main disease parameter needed to initialize the model is the transmission rate (β). The parameter β can be physically understood as follows: assuming a homogeneously mixed population, a randomly chosen susceptible person has c contacts per unit time with other persons, and p is the probability of disease transmission from an infected person to a susceptible person. Therefore, the quantity β is defined by the relationship: β = −c ln (1 − p) [51]. For sufficiently small p, β = cp. Thus, β is the product of contact rate and disease transmission probability.

The epidemic rate depends on several factors. Mitigation measures are mainly aimed at restricting the number of contacts between contagious and susceptible people, as well as the probability of infection by contact. Factors affecting incidence include migration, quarantines, lockdowns, heterogeneous mixing of the population, insufficient testing, the presence of asymptomatic cases, the simultaneous presence of several strains, etc. Regular adherence to restrictive measures based on population segments is difficult to assess. In reality, however, the dynamics of infection differ from the classical epidemiological process. Transmission and contact rates are functions of time and have a stochastic character (Figure 3 shows a portion of the daily number of COVID-19 cases in Karaganda from 10 March 2020 to 19 April 2020).

Therefore, many authors set the contact rate using the function of a probability distribution. Various studies have used normal distribution [52], lognormal [53], Bernoulli distribution [54], geometric distribution [55], and binomial distribution [56] in modeling. We defined the probability of contact as follows:$$p\left(c=x\right)=Geometric(x,\frac{1}{1+c})$$
where average daily contacts c = 6 [57].

To evaluate the parameter p, disease transmission probability, we used the OptQuest optimizer included in AnyLogic software. The difference between modeled and actual COVID-19 cases was chosen as the target function. The optimization consisted of minimizing the target function. As a result, different values of p were obtained during various pandemic periods.

One more parameter is the incubation period. In the literature, data on the incubation period varied widely from 1 to 14 days [49,50]; the average value was 5.8 days (95% CI from 5.0 to 6.7) [58]. According to a review that included 53 articles, the pooled mean incubation period for COVID-19 was 6.0 days (95% CI 5.6–6.5) globally [59]. We set an incubation period of 6 days.

### 2.3. Data Collection

We analyzed COVID-19 cases between 10 March 2020 and 16 November 2020 (250 days). Data were collected from the information system of the Ministry of Health of the Republic of Kazakhstan (Figure 4). The following procedure has been established in Kazakhstan: Interested accredited subjects of scientific activity should make a written request to the Ministry of Health of the RK. In the event of a positive decision, the request is redirected to the Republican eHealth Center, which prepares all the necessary information.

It was important for us to consider the possibility of forecasting when the real rate of disease is constantly changing, sometimes increasing and sometimes decreasing. We chose this time interval between 10 March 2020 and 16 November 2020 (250 days) because the total cases curve contains intervals of relatively constant growth and inflection points where the slew rate changes.

The first-time frame (AB) was 50 days, from 10 March 2020 to 30 April 2020. The second time frame (BC) was 50 days, from 01 May to 19 June. The third time frame (CD) was 20 days, from 20 June to 09 July. DE—30 days, from 10 July to 08 August; EF—50 days, from 09 August to 27 September; and FG—50 days, from 28 September to 16 November 2020.

For each of these time frames, we used the optimization mode in AnyLogic, and the optimal value of p, disease transmission probability, was selected. To observe the effect of stochastic variation on contact rate, we also used Monte Carlo simulation methods, with each run of the model containing 50 iterations. In each iteration, we used a different value from within the defined range for the contact rate. This action produced output values that were value ranges rather than point estimates. Next, we determined the mean values for each run.

To validate an agent-based model in each time frame, we compared the model output to real data. Based on each of these time periods, total cases were predicted for the next 50 days. We used the mean absolute percentage error (MAPE) metric for comparing simulated and true values. MAPE was chosen for its scale independence property, which removes bias due to test data size from the models. It is calculated by normalizing the average error at each point.
$$MAPE=\frac{1}{n}\sum_{t=1}^{n}\left|\frac{A_{t}-F_{t}}{A_{t}}\right|$$
where *A*_t_ is the actual data; *F*_t_ is the simulated data at time *t*; and n is the number of days.

Model accuracy was evaluated as highly accurate (MAPE% < 10), good (MAPE%: 10–20), reasonable (MAPE%: 20–50), and inaccurate forecasting (MAPE% > 50) [60].

### 2.4. Scenario Setting

Six different scenarios were developed, five of which reflect specific social interventions. The goal of the interventions in the model is to reduce the transmission rate (β). This can be achieved by reducing the likelihood of infection at each contact, for example, by wearing masks, maintaining physical distance, and reducing the number of contacts at home, school, work, or in the community.

#### 2.4.1. Base Scenario (BS)—No Intervention

The base scenario simulates a completely natural epidemic transmission process without any intervention measures. For the base scenario, we estimated parameter p according to the basic reproduction number R0, which is defined as the average number of secondary cases produced by a typical primary case during its infectious period in a susceptible population [61]. Parameter p can be inferred based on the mathematical expression R0 = cpDi, where Di is the average infectious period [62]. Therefore, the parameter p can be inferred as follows:p = R0/cDi

We set R0 = 3 and Di = 6 days. We determined that the patient is contagious during the incubation period before the onset of symptoms, after which they are immediately isolated [63].

#### 2.4.2. Scenario 2—School Closure (SC)

This scenario includes the closure of schools, preschools, and universities. The aim was to reduce social contact and cut transmission chains in the community.

#### 2.4.3. Scenario 3—Mask-Wearing (MW)

In practice, 100% mask-wearing is not achievable. Tian et al. [64] developed a simple transmission model that incorporated mask-wearing and mask efficacy as factors in the model. They found that wearing masks reduces R0 by a factor (1-mp)2, where m is the efficacy of trapping viral particles inside the mask and p is the percentage of the population that wears masks. In practice, 100% mask-wearing is not achievable. We considered a scenario with 50% mask usage and 50% mask efficacy [65]; then, (1-mp)2 = 0.56.

#### 2.4.4. Scenario 4—Vaccination (VS)

In this scenario, we assumed that the population was vaccinated with Gam-COVID-Vac at an average rate of 0.003%, which corresponds to the actual vaccination rate in Kazakhstan [66].

#### 2.4.5. Scenario 5—Combined Measures (CM)

This scenario includes mask-wearing, school closures, and vaccination.

#### 2.4.6. Scenario 6—Real Situation Simulation (RS)

The real situation included a set of intervention measures: closure of schools, mask regime, remote work, restriction of attendance at public events, and vaccination. However, we cannot assess the strictness of compliance with these measures.

For each scenario, the following outcomes were obtained:The number of infected people at home, in educational institutions, at work, in transport, and in stores.The number of infected people in different age groups.The required number of doctors, auxiliary medical, and technical personnel.The number of beds required for hospitalized persons.The cost of devices, medicines, chemicals, and other materials;

We used the COVID-19 Essential Supplies Forecasting Tool (COVID-ESFT, version 2.0) to generate a forecast model of the healthcare resources needed [67].

## 3. Results

### 3.1. Model Validation

During the fitting step in the optimize mode of AnyLogic, we specified the fixed seed value. 

In the Projects view, select the experiment.In the Properties view, open the Randomness section.Select the option Fixed seed (reproducible model runs).

In this case, a random number generator was initialized with the same value for each model run, and the model runs were reproducible. Table 3 presents the optimal values of p, disease transmission probability, for each time interval.

Then, we created a Monte Carlo experiment with 50 iterations. Figure 5 shows the fitting results in the form of an average value for all iterations and real data.

At the initial moment of modeling, the largest error between the model and real data was observed in the AB time frame. Several reasons can explain this difference, the first being mathematical. At the beginning of the COVID-19 pandemic, with a small number of cases, small differences created a large error rate. Another reason was possible overdiagnosis in the early days of the spread of the disease when diagnostic tests were not yet developed, and diagnoses were made according to the clinical picture. Increased alertness was also observed among doctors. The third reason may be the inaccurate duration of incubation periods since there are various data on this parameter in the literature [58]. In other time frames, good and high accuracy were obtained. In general, the model’s accuracy rate was 72%.

### 3.2. Prediction Simulation

Some timeframes in Figure 4 were chosen for training, whereas other parameters remained unchanged. We created a forecast for 50 days.

#### 3.2.1. Forecasting Based on Time Frame AB

Based on the first training time frame (from days 1 to 50), total cases were predicted using the Monte Carlo procedure (red line, Figure 6a). The predicted data were compared with the actual data (green line). Figure 6b shows forecasting accuracy (MAPE) for each of these days.

To uncover how the MAPE (%) changes with an increase in the duration of the forecast, Table 4 presents the results of forecasting every 5 days.

As already noted in this section, a significant difference was obtained between real and model data at the fitting stage. This difference was also reflected in the forecasting results. In this test time frame, the growth of the model’s total cases was exponential, whereas there was a linear trend in real data (Figure 6a). Accordingly, prediction accuracy also decreased over time. Therefore, if there was good and reasonable forecasting on the 5th and 10th days, the 15th day would be inaccurate.

#### 3.2.2. Forecasting Based on the AC Time Frame

In this case, the AC segment was used as the training time frame, and the prediction was calculated from the CE segment. On approximately the 126th day from the beginning of the observation, an inflection point (ε’) was observed on the curve when the growth rate of total cases slowed down (Figure 6c).

From this point onward, forecasting and real curves had different dynamics. In this case, according to MAPE, forecasting with an acceptable accuracy of < 50% was possible over the entire 50-day segment (Figure 6d). However, in the last 10 days, the relative prediction error (APE%) ranged from 52 to 116%.

In the time frames EF and FG, the total case curve was relatively linear. Therefore, prediction accuracy based on the training time frames AE and AF was higher than in previous cases. Therefore, when forecasting from 151 to 200 days, MAPE varied within 24–26%, which can be considered a “good” level of prediction (Figure 7a,b). In the period from 201 to 250 days, the accuracy was 95–97%, i.e., a high level (Figure 7c,d).

### 3.3. Epidemiological Impact of COVID-19

In this section, we considered various hypothetical scenarios for the spread of the disease. Our goal was to identify public places with the potential for intense transmission of the infection. We also aimed to identify the social groups most susceptible to infection. We wanted to assess the effectiveness of the local and central executive bodies’ protective measures in terms of public health and resource costs.

In the absence of any control measures (base scenario), we expected that the peak of infection would occur around day 58 of the pandemic and that the number of infected on that day would be about 140,000, or 28% of the population (Figure 8). Afterward, their number would decline and reach zero on day 130. We predicted that at this point in time, 471,746 (94%) of Karaganda residents would have been ill.

Interventions were able to slow or stop the spread of COVID-19. According to the SC scenario, the peak incidence would be reached somewhat later on day 72, and the pandemic would last 145 days. Vaccinating reduced and delayed the peak infection to 71,273 cases on day 66 and mask-wearing to 17,701 on day 182. According to the SC scenario, total cases would decrease by 7% compared to the baseline scenario, 54% according to the MV scenario, and 35% according to the VS scenario. Combining these methods (CM scenario) produced an effect of 99%. In reality, the total cases amounted to 145,044, which corresponds to a 70% decrease compared to the base scenario (Appendix A).

Computer simulation showed that in the base scenario, the highest number of infections occur at home (43%), then at work (29%), in transport (16%), in educational institutions (9%), and in stores (3%). In our model, stores are within walking distance of the house and are frequented by older people. For other scenarios, a similar ratio was observed (Figure 9a, Appendix A).

According to published data, the most severe outcomes of COVID-19 were in elderly patients, including the highest mortality rates, whereas younger individuals, especially children aged 1–18, were much more likely to display mild symptoms, if at all [68,69,70]. This trend, however, does not imply that older people play a leading role in the spread of COVID-19. Understanding the role of age in transmission and disease severity is critical for determining the likely impact of social-distancing interventions on infection transmission and estimating the expected global disease burden.

According to the BS, SC, MV, and VS modeling results, the maximum number of COVID-19 cases was observed in the 26–59 age group, i.e., among working people (Figure 9b, Appendix A). The proportion of this age group ranges from 62 to 78%. Then, the model suggested that 10–18% of all infections occurred in those aged 0–17 years. The minimum number of cases was among young people 18–25 years old (3–7%) and among the elderly ≥ 60 years old (9–12%). The model could not reproduce the observed age distribution of cases; the number of cases in children was overestimated and underestimated in older adults. In reality, the proportion of older people was 47%, and that of children was 8%. This does not mean that the elderly will necessarily play a leading role in the spread of COVID-19”. The reason for this discrepancy may be one of the shortcomings of our model, in which we did not account for the different susceptibilities to infection of children, adults, and the elderly. The parameter transmission rate may be related to age.

If we evaluate the effectiveness of the measures taken individually, mask-wearing takes first place. However, of course, preference should be given to a set of measures with the lowest incidence of disease.

### 3.4. Economic Impact of COVID-19 on Medical Resources

In this section, we assess the cost-effectiveness of preventative measures implemented by public health authorities. We compared the real costs of human and material resources with costs in other hypothetical scenarios. We used the COVID-19 Essential Supplies Forecasting Tool (COVID-ESFT, version 2.0) [67] to generate a forecast model of the healthcare resources needed.

Table 5 shows that the best way to reduce the burden of COVID-19 is to combine the protections listed above. Theoretically, only 159 persons would be hospitalized in such a scenario. The real situation was not so optimistic, but the simulation results showed that the maximum compliance requirements of the regulatory authorities significantly improved the epidemiological situation compared to the baseline scenario. Thus, closing schools reduces the number of hospitalizations by 7%, the mask-wearing mode by 54%, and vaccinations by 35%. In reality, the number of hospitalized people was 37,287, which is 70% less than in the baseline scenario.

A similar effect was observed in intensive care and artificial ventilation units.

#### 3.4.1. Human Health Resources (Medical Practitioners, including Physicians, Nursing Professionals, and Paramedical Practitioners)

One of many critical issues during the pandemic was staffing during the COVID-19 response. According to our modeling, all scenarios require the involvement of all 1970 hospital medical workers out of a total of 5325 medical workers in Karaganda. The ideal scenario would be strict compliance with a whole set of measures by all people. As for other categories of workers, according to the mask-wearing scenario, a decrease in the need for cleaners was predicted to be 19%, whereas, in reality, this decrease was 46% (Table 6).

Similarly, implemented measures significantly reduced the need for ambulance personnel. If 4458 workers were predicted in the base scenario, then in the mask-wearing mode, it would be 3611. In reality, the demand was for 2412 workers. Our calculations show that the need for technical staff was also significantly lower in reality than in the first four scenarios.

#### 3.4.2. Hospital Beds

A hospital bed is a bed specially designed for hospitalized patients or others in need of some form of healthcare. These beds have special features, both for the comfort and well-being of the patient and for the convenience of healthcare workers. The total bed capacity of the city’s hospitals is 7430, of which 4256 are for severely ill patients and 202 for patients in critical condition. During the pandemic, it became necessary to increase the number of beds (Appendix A). Infectious disease departments in all hospitals were expanded, and additional wards were opened for this purpose. At the peak of the pandemic, 3482 beds were deployed for patients with COVID-19, of which 214 beds were in intensive care units and 47 beds were in mechanical ventilation units. According to our data, the greatest need for additional beds occurred in intensive care and artificial ventilation units.

According to the simulation results for the basic scenario (Figure 10a), we predicted a shortage of 3198 beds (75% of available beds) for severe patients and 6496 for critical cases, requiring a 32-times increase in their number. Intervention measures can help alleviate the situation with several hospital beds. According to the VS, the need for additional beds is 601 for severely ill patients and 4162 for critical cases. There will be a shortage of 2861 beds in the MW for critical cases only. For the current number of patients, the deficit is 1857 beds for critical patients, which is 9.2 times more than the existing fund.

#### 3.4.3. Equipment, Pharmaceuticals, Consumables, and Accessories

Table 7 shows the cost of goods according to the price list on the site [67]. Base scenario costs are projected at USD 45,174,975. School closures and vaccination scenarios do not reduce the cost of equipment and supplies. In the mask-wearing mode, costs are reduced by 17% to USD 37,487,881, whereas in reality, they are USD 27,008,080, which saves 40% of costs compared to the base scenario (Figure 10b).

## 4. Discussion

### 4.1. The Findings and Their Implications

The COVID-19 pandemic has caused serious social and economic problems in sectors of the economy worldwide. The American Hospital Association estimated the monthly loss in lost productivity at USD 50.7 billion. For low- and middle-income countries, supporting effective health interventions is even more costly. The global problem was the unpreparedness of health facilities for this surge in morbidity. There were shortages of items such as personal protective equipment for health workers, hospital equipment, disinfectants, toilet paper, and water [71]. Kazakhstan, similar to other states, lacks the experience of rapidly spreading infections and their severe complications and high mortality. The pandemic proceeded differently depending on countries’ geographical, economic, and demographic situations. Many other factors, about which there are many publications in the scientific literature, also had an influence. Many restrictive measures were implemented in Kazakhstan for the first COVID-19 patients, which helped this period pass with relatively minor losses. Currently, it is time to reflect on the previous three years, evaluate the effectiveness of the strategies used to fight the pandemic and prepare a scientific basis for predicting similar epidemiological situations.

In this study, we aimed to assess the social and economic consequences of the pandemic under the conditions of preventative measures by state regulatory authorities. For this purpose, we used two tools: an agent-based model of COVID-19 spread in Karaganda (Kazakhstan) and the WHO tool for material and labor cost estimation. We preferred the ABM to the population-based model because it allows us to take into account the social activity of different segments of the population and thus estimate morbidity among children, adults, and the elderly to model the impact of preventative measures in each social group, and to plan material and human resources at the level of hospitals and polyclinics. We set out to create a model that realistically represents the daily activities of different populations under uncertainty. These uncertainties include adherence to restrictive measures, susceptibility to infection, susceptibility to vaccines, duration of illness, influence of environmental factors, etc. Therefore, we developed a stochastic ABM.

One of the tasks that we set ourselves was forecasting based on the created model. Earlier attempts were made in Kazakhstan to forecast the situation with the spread of COVID-19. However, time has shown that the forecast data were greatly overestimated compared to the real data. Therefore, in [72], it was projected that within six months, the number of infected individuals in Kazakhstan would be 982,010, but, in reality, it was 11,239. In another paper [73], a 120-day prediction of all exposed and infected individuals will give a total of 188.983, which is 8.5 times higher than the real data. In their work, the authors used SEIR models based on differential equations. The main reasons for unsuccessful predictions are insufficient data on the nature of the spread of infection in the early stages of the pandemic and the impossibility of validating the model for the same reason. It is now evident that significant modification of classical models or application of other types of models is required to analyze COVID-19.

In our ABM model, we used total cases as a predictor of the spread of COVID-19. Satisfactory results were obtained with long-term (up to 50 days) forecasting due to monotonous changes in this indicator. However, developing events have shown that the dynamics of daily incidence are oscillatory, and periods of relative stability are accompanied by sudden outbreaks, the nature of which has not yet been explained. As a result, inflection points appear on the total case curve. Two such points can be identified in Figure 4. In this case, relatively satisfactory results were obtained for short-term forecasting up to 10 days.

The spread of infection largely depends on the behavior of various social groups, their adherence to precautionary measures, age, health, mentality, marital status, and other factors. The most socially active are young people, and more contacts occur in this group. In some countries, many elderly people are more susceptible to severe forms of the disease, resulting in prolonged treatment. Knowledge of such features in advance can help prepare for various scenarios. Simulation of various scenarios shows, as expected, that the most important places of transmission of infection are families and public transport [74,75]. The city of Karaganda has more than 173 thousand households, with 35% of families consisting of four or more people. We observed a very high risk of domestic infection in 50% of homes. Using various preventative measures in such cases did not show any improvement. Although mask-wearing effectively blocks close-contact transmission, people do not always wear masks in places they frequent daily (e.g., home, office), even when they show symptoms of illness. There does not appear to be any real means of preventing the spread of infection in the household other than herd immunity.

Another common place for spreading infection is public transport (buses are the only type in cities) because the availability of personal transportation in the city is only 20%. Reducing the number of bus routes can also be an effective strategy.

We also investigated age dependence in clinical cases. A higher number of patients were aged 18–59. Children and youth, and then the elderly, comprised about 12% of total cases. These results diverge from real data. In fact, the proportion of people aged 60+ was 47%. This discrepancy can be explained by the fact that our simplified model did not consider children as less susceptible than adults to infection by contact with an infectious person, decreasing cases among children. Children often develop the disease asymptomatically but remain a source of infection. At the same time, older people are more susceptible to infection; they often have chronic illnesses, and their immune systems are weakened.

We analyzed the efficiency of several main interventions, such as school closures, mask-wearing, vaccination, and their combination. According to our data, these methods have different efficiency. Thus, we predicted that the most effective was mask-wearing, which could reduce incidence by 2.2 times. Another measure was the closure of educational institutions. In the literature, there is evidence that school closures are an ineffective measure of protecting against the spread of infection. Some studies reported that school closures were associated with no change in transmission. As noted above, children may be less likely to transmit COVID-19 than adults, resulting in limited transmission in schools as well as from schoolchildren to adults. Our results are consistent with these findings. A simulation of this scenario showed a 7% reduction in incidence.

Vaccination is known to help reduce the risk of infection, but it is not significantly effective [76]. In our model, vaccination reduced the incidence by 35%. Clearly, vaccination should prioritize high-risk workers in healthcare, basic services, food processing, and transport, as well as people > 60. Ideally, combining these measures and strict adherence to all regulations could minimize damage from the pandemic. Of course, reality is much richer than any model, regardless of how complex it is. Kazakhstan’s state bodies used all the available tools to prevent the spread of infection. As a result, the number of cases in Karaganda was 3.2 times less than in the baseline scenario. Mobilizing human resources to fight the pandemic helped avoid a shortage of doctors and technical staff, significantly reducing their needs compared to the baseline scenario.

The number of hospital beds generally corresponded to the needs of the situation. The shortage was felt in beds for severe and critical patients, which is understandable since hospitals were unprepared for the large influx of patients. This problem was solved by reorienting some non-infectious departments to combat COVID-19 and opening provisional centers.

We also determined the economic impact based on lower cash costs for medicines, reagents, medical devices, and equipment compared to the hypothetical “do nothing” scenario. Even a simple mask-wearing approach could reduce material costs by 17%. According to our forecast, the savings amount to 40%.

Clearly, the economic cost of not containing the virus would be unimaginably high. However, the cost of mitigation measures is still a major challenge for the government and other stakeholders. It appears that a complete lockdown is an unnecessary measure. Experience shows that it is fraught with severe post-pandemic social and economic consequences [19]. Therefore, governments must find the optimal balance of freedoms and restrictions while being prepared for various scenarios.

### 4.2. Limitation

This study had some limitations. Our model could not reflect the diversity of human activity inherent in real life. For simplicity, we assumed that people travel to at most one place each day (for students and workers, public transport is an intermediate link) and spend a certain amount of time there, which varies between places and age groups. The scenarios considered were idealized; for example, schools allowed special classes for graduates, vaccination rates were uneven, and mask compliance was difficult to assess. All of this may lead to underestimations of the number of agent contacts and the rate of infection.

Some model parameters derived from previously published studies varied across studies. For example, we adopted the incubation period as 6 days. However, different authors have suggested an incubation period of COVID-19 from 3 to 11 days. A longer incubation period leads to a later onset of symptoms and the isolation of the patient, which in turn increases the number of people in contact with him. Therefore, we averaged the data to avoid potential inaccuracies in our model. In our model, we assumed that after symptoms of the disease appear, people are immediately isolated, and those who have recovered from COVID-19 cannot be reinfected throughout the entire epidemic period. However, evidence to support this assumption is limited.

Moreover, we did not take into account that different segments of the population have different susceptibilities to infection. As a result, we received underestimated morbidity data among older people and overestimated data among young people. Nevertheless, these shortcomings do not detract from the significance of the results obtained. Despite these limitations, our model matched real data very closely, as was shown in Section 3.1.

At the same time, these limitations provide a basis for further refinement of our model. In the future, it should reflect the stochasticity of parameters such as incubation period, daily number of vaccinated persons, number of adherents to preventative measures, etc. The possibility of contact between different social groups in educational institutions, in shops, and in healthcare centers should also be considered. It is also necessary to specify the susceptibility of certain social groups to infection. All this will increase the predictive power of the model.

## 5. Conclusions

At the beginning of the pandemic, there was a strong request from the Ministry of Health to forecast the future situation of COVID-19 in Kazakhstan, as no one could imagine what resources would be needed to fight the pandemic. The Republic had no experience in building such models, especially in determining the epidemiological parameters of the model. Attempts to use ready-made tools proposed by various authors as events unfolded resulted in greatly overestimated predictions. Most of the predictions were made in the early stages of the pandemic when the number of cases was increasing rapidly. If there is an inflection point on the curve of the total number of cases, the predictions deviate significantly from reality. Therefore, we believe that our model is the basis for the creation of a real working model that takes into account the peculiarities of the social behavior of the inhabitants of the Republic and the peculiarities of the structure of the healthcare system. We cannot rule out the possibility that, in the future, humanity will again encounter viruses such as SARS-CoV-2. In that case, we must be ready to respond quickly to public health needs.

Long-term forecasting (up to 50 days) has produced satisfactory results for monotonic changes in this indicator. However, when inflection points appear on the “total number of cases” curve, relatively satisfactory results were obtained for short-term forecasting up to 10 days.

The measures taken by the state authorities of Kazakhstan were primarily aimed at saving people’s lives. In our opinion, they were quite effective since the incidence of COVID-19 in the Republic was 8000/100,000 population, and the mortality rate was 101/100,000 population, which is much lower than in many countries.

In the future, in similar situations, the combined tool we have proposed will provide scientifically based information on the possible development of the infectious process to be obtained and make it possible to provide health organizations with specialists and material resources in a timely manner.

## Figures and Tables

**Figure 1 healthcare-11-02968-f001:**
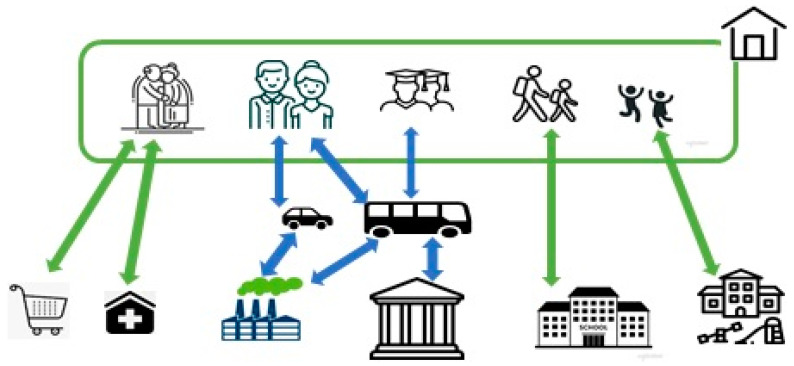
Illustration of a simplified model of multilayer catenary networks in the model. Individuals move between home, school, work, and the community throughout the day.

**Figure 2 healthcare-11-02968-f002:**
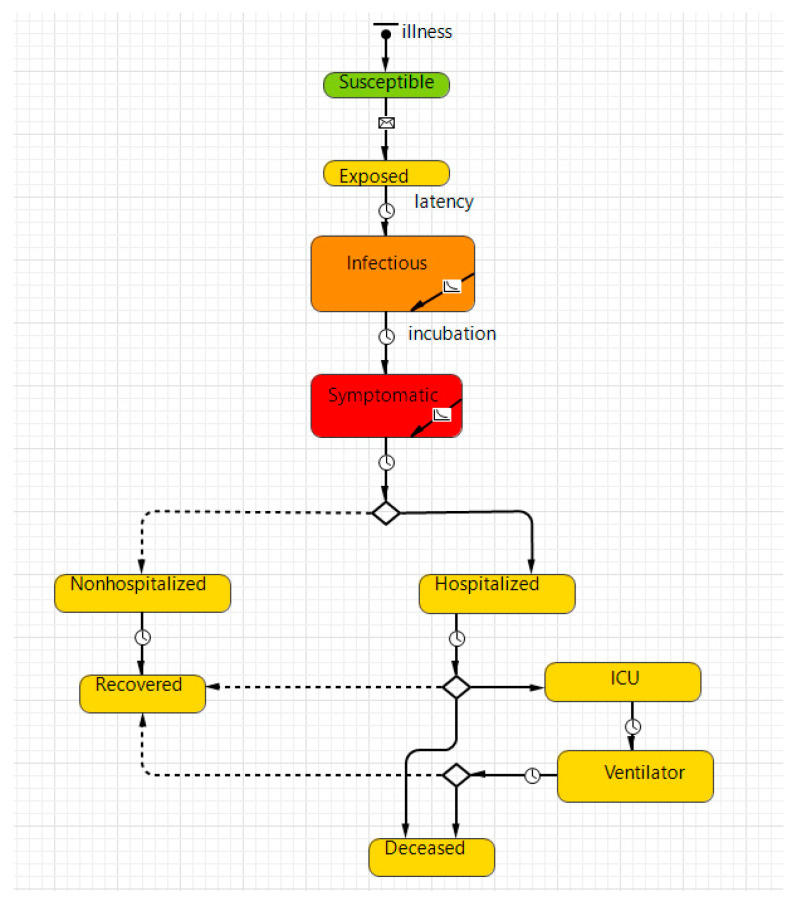
Disease transmission.

**Figure 3 healthcare-11-02968-f003:**
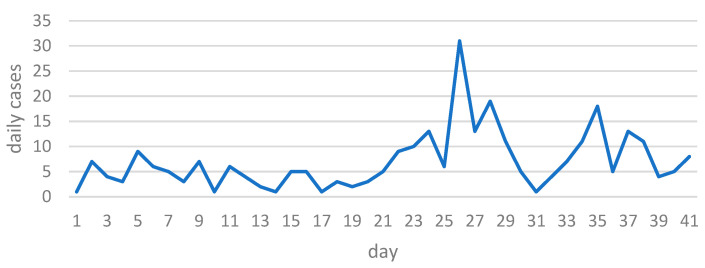
Daily number of COVID-19 cases from 10 March 2020 to 19 April 2020.

**Figure 4 healthcare-11-02968-f004:**
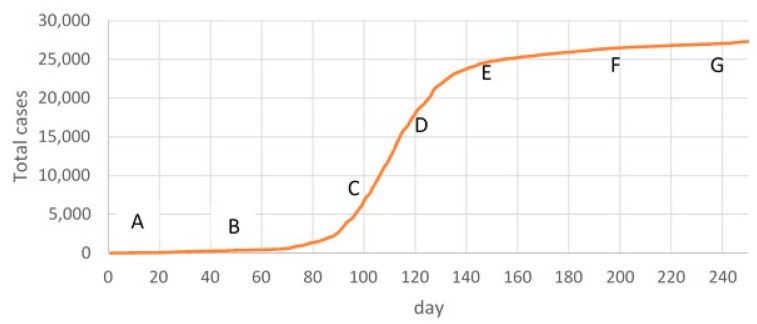
Total cases of COVID-19.

**Figure 5 healthcare-11-02968-f005:**
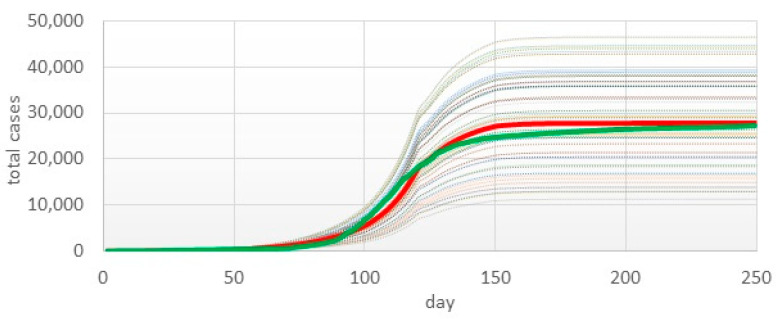
Parameter fitting result. Red line—average value over 50 iterations; green line—real data.

**Figure 6 healthcare-11-02968-f006:**
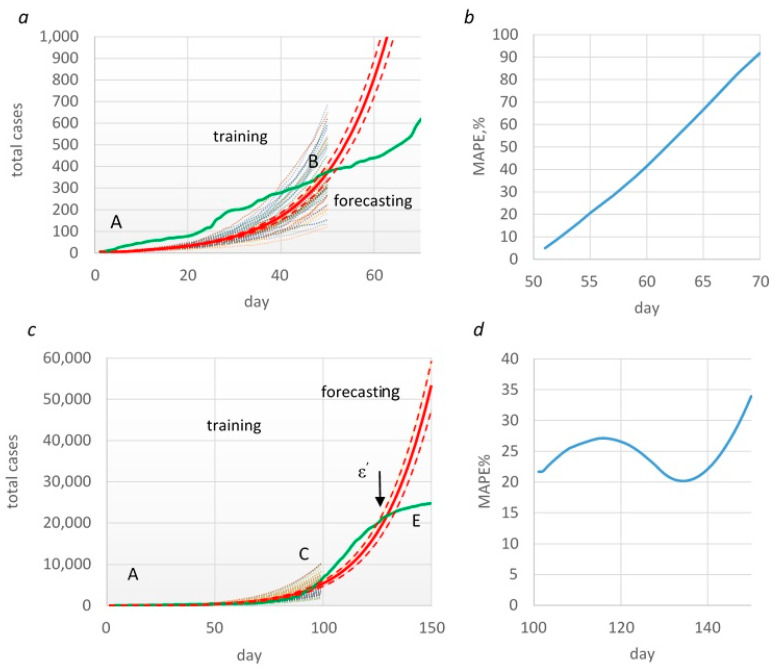
Forecasting total cases and evaluation of forecasting accuracy in AB time frame (**a**,**b**) and in AC time frame (**c**,**d**). Green line—real data; red line—mean (95% CI) predicted data.

**Figure 7 healthcare-11-02968-f007:**
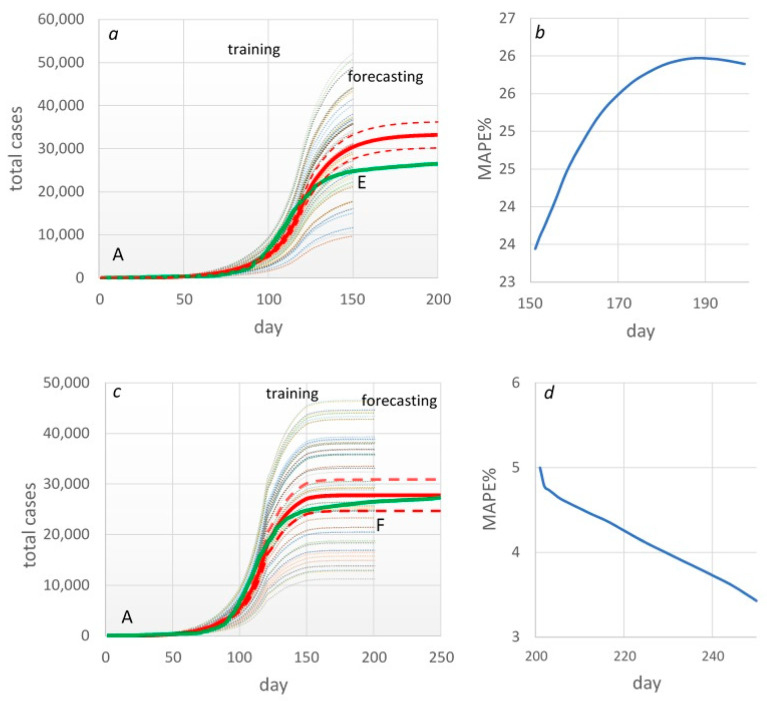
Forecasting total cases and evaluation of forecasting accuracy MAPE in AE (**a**,**b**) and AF (**c**,**d**) time frames. Green line—real data; red line—mean (95%CI) predicted data.

**Figure 8 healthcare-11-02968-f008:**
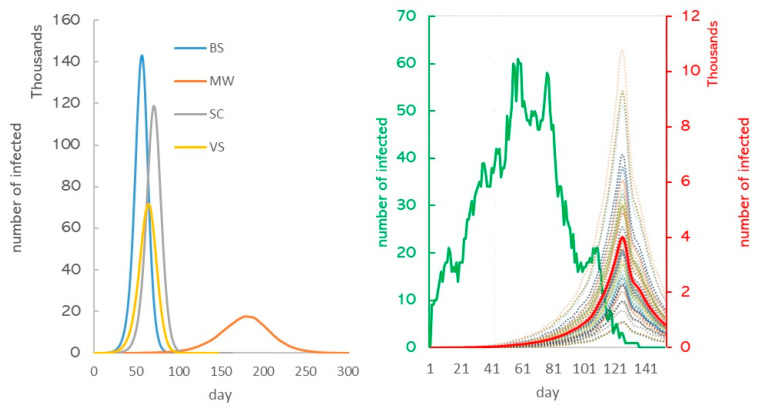
Estimated epidemic curves of the COVID-19 outbreak under various scenarios. BS: base scenario; MW: mask-wearing; SC: school closure; VS: vaccination; CM: combined measures; RS: real situation.

**Figure 9 healthcare-11-02968-f009:**
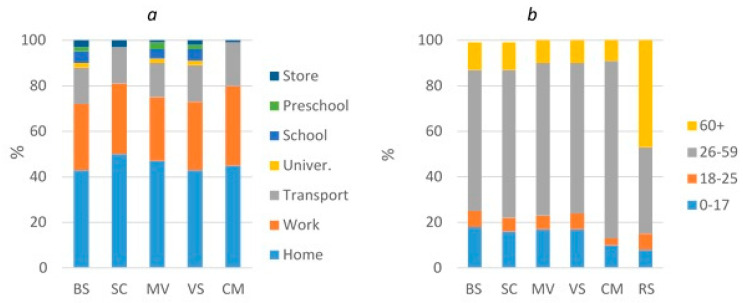
Infection by location (**a**) and age group (**b**).

**Figure 10 healthcare-11-02968-f010:**
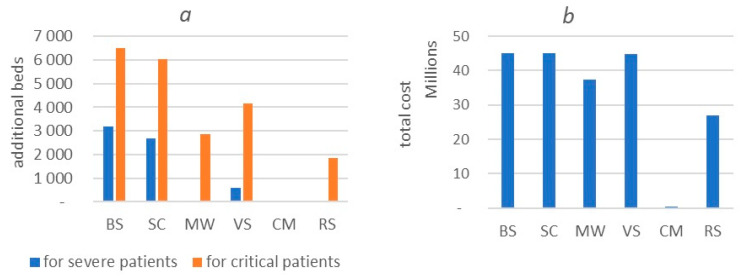
Number of additional beds needed (**a**) and financial costs for equipment and materials (**b**). BS: base scenario; MW: mask-wearing; SC: school closure; VS: vaccination; CM: combined measures; RS: real situation.

**Table 1 healthcare-11-02968-t001:** Social infrastructure of Karaganda.

Variable	Number	References
Population	500,896	https://stat.gov.kz (accessed on 1 July 2023)
Total families	173,501	
Enterprises	14,418	
Schools	87	
Kindergartens	112	
Universities	7	
Hospitals	14	
Outpatient clinics	7	
Grocery shops	1488	
Public transport (buses)	540	
Private vehicle	105,230	

**Table 2 healthcare-11-02968-t002:** Epidemiological parameters.

Variable	Unit	Current value	References
Average daily contact	number	6	[49]
Average incubation period	day	6	[50]
Adult hospitalization rate	%	29	Ministry of Health
Children hospitalization rate	%	8,1	
Case severity			
- Mild	%	74	
- Moderate	%	23	
- Severe	%	2	
- Critical (inpatient, ventilation)	%	1	
Fatality rate	%	1,1	
Critical fatality rate	%	91	
Duration of outpatient treatment	day	triangular (8,10,14)	
Duration of inpatient treatment	day	triangular (15,20,28)	
Duration of critical care (invasively ventilated)	day	triangular (9,10,14)	

**Table 3 healthcare-11-02968-t003:** Model accuracy.

Table	*p*-Disease Transmission Probability	MAPE%
0–50	0.09	53
50–100	0.069	41
100–120	0.084	17
120–150	0.024	5
150–200	0.011	8
Overall		28

**Table 4 healthcare-11-02968-t004:** Accuracy of forecasting from 51 to 65 days.

Forecasting Period	51–55 Days	56–60 Days	61–65 Days
MAPE%	20	42	65.7

**Table 5 healthcare-11-02968-t005:** Impact of COVID-19 on hospitalization in the simulated scenarios.

Scenario	Total Cases	Outpatient	Hospitalized	ICU	Ventilated
BS	471,746	349,092	122,654	7359	1651
SC	438,610	324,571	114,039	6842	1535
MV	215,715	159,629	56,086	3365	755
VS	307,381	227,462	79,919	4795	1076
CM	604	445	159	10	2
RS	145,044	107,757	37,287	2294	502

**Table 6 healthcare-11-02968-t006:** Healthcare workers demand in the simulated scenarios.

		BS	SC	MV	VS	CM	RS
Inpatient	Total number of healthcare workers	1970	1970	1970	1970	54	1970
	Total number of cleaners	3121	3121	2528	3121	13	1688
	Total number of ambulance personnel	4458	4458	3611	4458	18	2412
	Total number of biomedical engineers	134	134	108	134	1	72
Laboratories	Total number of lab staff required	3	3	3	3	3	3
	Total number of cleaners	1	1	1	1	1	1

**Table 7 healthcare-11-02968-t007:** The costs of goods (USD).

	BS	SC	MW	VS	CM	RS
Hygiene	325,498	305,174	163,439	224,683	665	109,787
Personal Protective Equipment	681,549	655,629	452,237	552,981	4204	351,771
Diagnostics	69,840	69,840	67,320	69,840	13,088	63,720
Pharmaceuticals	4,614,981	4,614,981	3,776,023	4,614,981	26,438	2,588,645
Biomedical Equipment and Non-consumables	39,417,714	39,417,714	32,972,178	39,417,714	370,944	23,849,801
Biomedical Consumables and Accessories	65,392	65,392	56,682	65,392	1015	44,356
Total	45,174,975	45,128,731	37,487,881	44,945,592	416,354	27,008,080

## Data Availability

The data presented in this study are available upon request from the respective author. Data accessibility requires permission from the Ministry of Health of the Republic of Kazakhstan.

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
