# Peer review of "The Epidemiological and Economic Impact of COVID-19 in Kazakhstan: An Agent-Based Modeling"

_healthcare, 2023, doi:10.3390/healthcare11222968_

Round 1

Reviewer 1 Report (Previous Reviewer 2)

Comments and Suggestions for Authors

The critique formulated for the original version of the article pointed out the limited discussion of examples from the research literature on the use of agent-based models for assessing and simulating the processes of the Covid-19 pandemic. The three exemplary publications suggested by the reviewer were utilized, but the authors did not conduct a review of their application by other subsequent authors. Literature reviews as well as comparative analyses with other authors still require additions. The presented individual case study for Kazakhstan is comprehensive, but comparative analyses with other examples enrich the research findings.

Author Response

Reviewer 2 Report (New Reviewer)

Comments and Suggestions for Authors

Dear authors,

See below my comments for improvement.

Abstract

This is a well-structured abstract, but there are some areas where clarity, brevity, and grammar can be improved. Here is a breakdown of my suggestions:

  1. Title & Affiliation The title and affiliations seem clear. However, for clarity, consider whether “Agent-Based Modeling” would be more precise as “Using Agent-Based Modeling” or “An Agent-Based Modeling Approach.”
  2. Abstract a. Background:
    • Slight grammar improvement: “Our study aimed to assess how effective the preventive measures taken by the state authorities during the pandemic were in terms of public health protection and rational use of material and human resources.”

b. Materials and methods:

    • Consider being more succinct: “We utilized a stochastic agent-based model for COVID-19 spread, combined with the WHO-recommended COVID-ESFT version 2.0 tool for material and labor cost estimation.”

c. Results:

    • Streamline for clarity and remove redundancy: “Our long-term forecasts (up to 50 days) showed satisfactory results with a steady trend in total cases. However, short-term forecasts (up to 10 days) were more accurate during relative stability interrupted by sudden outbreaks. Simulations indicated that infection spread was highest within families, with most COVID-19 cases in the 26–59 age group. Government interventions resulted in 3.2 times fewer cases in Karaganda city than predicted under a ‘no intervention’ scenario, yielding an estimated economic benefit of 40%.”

d. Conclusion:

    • Slight rephrasing for clarity: “The combined tool we propose can accurately forecast the progression of the infection, enabling health organizations to allocate specialist and material resources timely.”
  1. Keywords: They seem appropriate for the paper.
  2. General Suggestions: a. Avoid passive voice to make statements more direct and concise. b. Ensure consistent verb tenses throughout. c. Use precise language to clearly state findings, especially in the results section.

Introduction

The introduction provided is comprehensive, detailed, and well-structured. Here are some comments and suggestions to further improve its clarity and coherence:

  1. Topic Introduction (Lines 29-38):
    • The introduction effectively establishes the context by highlighting the impact of the COVID-19 pandemic on global healthcare systems and the importance of understanding disease dynamics.
    • The mention of Kazakhstan provides specificity, showing that the study will focus on a particular country’s experience.
  2. Background on Modeling (Lines 39-55):
    • The different types of models used historically (differential equations, cellular automata, deterministic models) to predict epidemics are summarized well.
    • However, consider introducing acronyms (like SI, SIR, SEIR) more thoroughly or providing a brief definition for readers unfamiliar with these models.
  3. Agent-Based Models (Lines 56-81):
    • This section thoroughly reviews agent-based models (ABMs) and their application in COVID-19.
    • It might be helpful to add a sentence or two explaining the unique advantages or differences of ABMs compared to earlier models for readers unfamiliar with the topic.
  4. Economic and Epidemiologic Impacts (Lines 82-97):
    • This section smoothly transitions from purely medical implications to the socio-economic impacts of the pandemic.
    • Using specific studies to illustrate the points made adds credibility to the narrative.
  5. Modelling Challenges (Lines 105-156):
    • The challenges faced in predicting the spread of COVID-19 using various models are described in detail. This section sets the stage for the importance of the author’s study.
    • Consider simplifying some technical explanations or adding a summary at the end for readers who might not have a deep understanding of modeling techniques.
  6. Kazakhstan-Specific Challenges (Lines 157-159):
    • This section connects the general modeling challenges back to Kazakhstan, reinforcing the focus of the study.
  7. Study Objectives (Lines 160-174):
    • The objectives and contributions of the study are listed. This provides a clear roadmap for readers to understand the rest of the paper.
    • Consider refining the wording in some places for clarity. For instance, “mask veering” (Line 174) might be a typo and should perhaps be “mask-wearing.”
  8. General Suggestions:
    • Citations: Ensure that all sources ([1], [2], etc.) correspond to the correct references in the bibliography.
    • Consistency: Ensure that all terms, acronyms, and abbreviations are consistently used throughout the introduction and the paper.
    • Simplification: While the introduction is comprehensive, it could benefit from slight simplification in parts to cater to a broader audience.
    • Transitions: Smooth transitions between sections can help readers follow the narrative more easily.
    • Proofreading: Lastly, ensure thorough proofreading to catch and correct any typographical or grammatical errors.

Material and methods

  1. Clarity and Precision:
    • Study Area: While it is informative to know the city’s significant characteristics, the relevance of some information, like the number of high schools or referral hospitals, to the spread of the disease could be made more explicit. Consider clarifying the relevance of these details to the study.
    • ABM Model Implementation: The purpose or relevance of mentioning “Chicago, United States” is unclear. If it is the software company’s location, this could be removed or placed in a footnote for clarity.
  2. Model Simplifications and Assumptions:
    • Explain your simplifications’ potential limitations or implications, especially in modeling social contacts. By understanding these, the reader can better gauge the reliability and applicability of your model to real-world scenarios.
  3. Parameter Definitions:
    • A table summarizing the key parameters used in the model, their definitions, and their sources/values would help readers quickly understand the model’s intricacies.
  4. Data Collection:
    • Expand on the methods used for data collection from the Ministry of Health. Was it manual data extraction, API usage, or some other way?
  5. Scenario Setting:
    • Justify why specific scenarios (like school closures, mask-wearing, etc.) were chosen over others. This can provide insights into the real-world relevance and applicability of your model.
    • Adding a brief conclusion or expected outcome after each scenario would be beneficial. This would give readers an idea of what to expect or look for in the results section.
  6. Validation:
    • You mention MAPE as a metric for validation. Consider also discussing other validation or calibration techniques, if any were used.
  7. Complexity:
    • Given the intricate details, consider providing a summary or overview at the start of the section. This “roadmap” will help guide readers through the complexities of the model.
  8. Formatting:
    • Ensure consistent formatting throughout the section. This includes consistently using bullet points, spacing, indentation, and subheadings.

Results

  1. Clarity and Terminology:
    • Ensure consistency in terminology, e.g., avoid using both “p-disease transmission probability” and “p-transmission probability.”
  2. Reproducibility:
    • It is mentioned that the same seed value was used in Optimize mode, making model runs reproducible. However, a transparent methodology on how this was achieved should be noted.
  3. Textual Repetitions and Redundancies:
    • The explanation about initializing the random number generator with the same value for reproducibility is repeated. Such repetitions should be avoided to keep the text concise.
  4. Address Discrepancies:
    • The results indicate discrepancies between the model’s predictions and real-world data, especially concerning the age distribution of cases. Discuss the possible reasons for this discrepancy and propose ways to improve the model’s accuracy.
  5. In-Depth Explanation:
    • The “Epidemiological impact of COVID-19” section discussed various hypothetical scenarios. Providing more detailed definitions or explanations for methods like SC, MV, VS, CM, etc, would be beneficial.
  6. Economic Impact:
    • When discussing the economic impact of COVID-19 on medical resources, more detailed financial data or calculations could be provided to give readers a clearer picture of the costs involved.
  7. Recommendations and Further Study:
    • The section could benefit from a subsection that proposes specific recommendations based on the results. For instance, if mask-wearing proves to be highly effective in reducing transmission in the simulations, this could be emphasized as a recommended intervention.
    • Propose areas for further study, mainly focusing on the limitations observed in the current model.
  8. Conclusion:
    • While this is the “Results” section, having a summary or Conclusion at the end might be beneficial, summarizing the main findings.
  1. Editing and Proofreading:
  • Some sentences are worded awkwardly or redundantly, like: “Several reasons can explain this difference can be explained...”. Proofreading and revising for clarity and brevity will enhance readability.
  1. Context:
  • Provide more apparent context or background about the geographical location or population under study, as terms like “the city’s residents” are mentioned without specifying which city.

Discussion and conclusions

Discussion:

  1. Structure and Flow: The discussion flows from the bigger picture of the pandemic, moves on to modeling, and then to specific results. It might be beneficial to maintain a more precise flow:
    • Start with the particular aims of the study.
    • Address the significant findings concerning the purposes.
    • Discuss these findings in the broader context of the pandemic.
  2. Comparative Analysis: When discussing Kazakhstan’s unique features and outcomes, compare with other countries to provide perspective.
  3. Additional Detail on Modeling: Elaborate on why stochastic agent modeling was chosen and how it compares with other modeling techniques regarding advantages and limitations.
  4. Expand on Uncertainties: You have mentioned several uncertainties related to the pandemic. Consider a dedicated subsection discussing these uncertainties and how they impacted your model’s outcomes.

Conclusions:

  1. Bullet Points: Bullet points or numbered lists can be helpful, but ensure they are concise. A paragraph format might serve better for in-depth explanations.
  2. Clarify Overestimated Predictions: When you mention that other tools resulted in overestimated predictions, providing examples or quantifying this difference would be helpful.
  3. Recommendations: While the conclusions reflect on the past, adding a set of recommendations based on the findings would be beneficial. For instance, what specific actions could be taken if another pandemic arises based on your model?

General Suggestions:

  1. Language and Tone: Ensure the language is neutral and unbiased. Avoid making overly definitive statements, especially when discussing the limitations or uncertainties of the study.
  2. Clarify Terms and Assumptions: Some terms like “bed fund” or “mask mode” may not be familiar to all readers. Clarify or provide context where necessary.
  3. Limitations Section: Consider detailing more about how each limitation impacted your results. This gives the reader an understanding of how to interpret the results in light of these limitations.
  4. Future Directions: In the conclusions or as a separate section, discuss potential future directions for this research. What next steps or related areas should be explored based on your findings?

Yours sincerely

Comments on the Quality of English Language

Dear authors,

Regarding the English, pay attention to the following:

Abstract:

  1. Improve clarity, brevity, and grammar.
  2. Refine the title for clarity.
  3. Rephrase specific sentences to enhance transparency and remove redundancy.

General Suggestions for Abstract:

  1. Avoid passive voice.
  2. Ensure consistent verb tenses.
  3. Use precise language, especially in the results section.

Introduction:

  1. Introduce acronyms more thoroughly.
  2. Add more information about agent-based models’ unique advantages.
  3. Simplify some technical explanations.
  4. Refine wording for clarity (e.g., “mask veering” might be a typo).
  5. Ensure correct citations.
  6. Maintain consistency in terms.
  7. Simplify some sections.
  8. Improve transitions.
  9. Proofread for typographical or grammatical errors.

Material and methods:

  1. Clarify the relevance of certain information.
  2. Mention the purpose of specific references like “Chicago, United States.”
  3. Explain the limitations of model simplifications.
  4. Provide a summary table of critical parameters.
  5. Elaborate on data collection methods.
  6. Justify the selection of specific scenarios.
  7. Discuss other validation techniques.
  8. Provide an overview at the start of the section.
  9. Ensure consistent formatting.

Results:

  1. Ensure consistent terminology.
  2. Elaborate on the methodology for reproducibility.
  3. Avoid textual repetitions.
  4. Discuss discrepancies between model predictions and real-world data.
  5. Provide detailed definitions or explanations for various scenarios.
  6. Provide more detailed financial data.
  7. Propose recommendations based on results.
  8. Include a summary or conclusion.
  9. Edit and proofread for clarity and brevity.
  10. Provide context for terms like “the city’s residents.”

Discussion and conclusionsDiscussion:

  1. Improve the structure and flow.
  2. Compare Kazakhstan’s outcomes with other countries.
  3. Elaborate on modeling choices.
  4. Expand on uncertainties.

Conclusions:

  1. Decide between bullet points or paragraph format.
  2. Clarify statements on overestimated predictions.
  3. Provide recommendations based on findings.

General Suggestions for Discussion and Conclusions:

  1. Ensure neutral and unbiased language.
  2. Clarify or provide context for specific terms.
  3. Detail the impact of each limitation.
  4. Suggest future directions for the research.

Yours sincerely

Author Response

Reviewer 3 Report (New Reviewer)

Comments and Suggestions for Authors

The proposed paper ‘The Epidemiological and Economic Impact of COVID‑19 in Kazakhstan: Agent-Based Modeling’ presents an advanced statistical modelling oriented to predict the number of infected people and the related resources necessary to treat them under different scenarios of pandemic mitigation policies, using as study case the city of Karaganda, Kazakhstan.

I want to express my appreciation for the scientific soundness of the work, and for the clear manuscript drafting. Nonetheless, I think there are some issues that authors should address to improve the paper.

GENERAL ISSUES:

1) The research context (as described in the introduction) is missing one important field, which is the application of AI (mainly machine learning) models in monitoring COVID-19. This branch is important because addresses two of the main limitations in most models, which are the reliability of data sources (by using proxy data such as those of emergency medical services, emergency department accesses etc.), and the confounding effects of additional socio-demographic and environmental factors. I think that including one paragraph about this research line (although a complete review is out of scope and unfeasible) could substantially increase the value of the manuscript.

2) In strict connection with the previous issue, also the geospatial dimension of the phenomenon is completely out of the context description. While it is not always primary, depending on the objectives, the different approaches tested in scientific literature (with a primary role of machine learning) to handle this aspect are definitely worth a mentioning in the introduction.

3) While a simplification of social groups and relative behaviors is indeed necessary, there is also a need to account for variability. Was some kind of ‘noise’ added to the model of social behaviors? If not, this may be reported as a limitation and possible study development.

4) What is the level of reliability of official diagnosis data? Considering that the analysis is referred to the first and second pandemic period, in many areas worldwide the availability of testing supplies was a strong limitation in testing capabilities. As reported in lines 357-359, this was an issue in the study case too. What was its impact across the different phases, and how was this accounted for?

5) Lines 440-445: The levels of diffusion of the diseases in the different age groups should be assessed in terms of incidence (or prevalence), i.e. number of cases / overall population in a certain time period (overall). Considering the age distribution of the occurred cases may not be enough representative.

6) At lines 336-337 you report <<COVID-19 Essential Supplies Forecasting Tool (COVID-ESFT, version 336 2.0) to generate a forecast model of the healthcare resources needed>> Was the whole analysis preformed in this environment? Seems to be only referred to results reported in section 3.4. Is the code/script available for reproducibility of all the different parts of the analysis framework?

7) As different solutions were adopted around the world, it is necessary to explicit what kind of vaccine (with official effectiveness data) was included in the model.

TEXT-SPECIFIC ISSUES:

8) Line 32: <<unprecedented measures to support the industry, which significantly reduced the spread of infection>> If measures were undertaken to reduce the spread of the pandemic, I guess they were restricting people’s movements, probably also in terms of working activities. So it is counterintuitive why these measures are considered to be supporting the industry, at least in the short-term.

9) Line 33: I guess now it is better to say ‘recent years’ instead of ‘recent months’.

10) Line 45: Also models based on AI (specifically machine learning) should be included in the new approaches.

11) Line 195: figure 1 requires some additional text (both in the figure and in the caption), as it is a little unclear in the current form.

12) Line 249: please indicate what is the time period to which data reported in figure 3 are referred to.

13) Lines 277-281: What is the rationale for this temporal subdivision?

14) Line 328: what are the specifics of scenario 6?

15) Line 354: repetition typo

16) Line 378: it is a little unclear what are the time intervals reported: captions says 51-65 days, labels are 5-15.

17) Line 417: figure 8 needs a legend. Same for scenario in figure 10.

18) Line 432: figure 9 is a little too low on resolution

19) Line 470-471: <<According to our simulation, implementing various preventative measures did not decrease the total number of workers fighting the disease.>> Considering the wide range of infection cases in the different scenarios, this seems a little bit odd. Can you elaborate on this?

20) Lines 480-482: <<Calculations show that the need for technical staff was also significantly lower in reality than in the first four scenarios.>> This is also somehow odd. Can this be related to the overload of workers that likely occurred in the real scenario? Shouldn’t this be taken into account?

21) Line 549: I am missing the point about how reducing bus routes should help decreasing the infections. Wouldn’t this result in more people stacking on the same bus?

22) Line 586-591: some references are recommendable for the different statements.

Author Response

This manuscript is a resubmission of an earlier submission. The following is a list of the peer review reports and author responses from that submission.

Round 1

Reviewer 1 Report

Comments and Suggestions for Authors

Review report healthcare-2609609

THE EPIDEMIOLOGICAL AND ECONOMIC IMPACT OF COVID‑19 IN KAZAKHSTAN.

AGENT BASED MODELLING

Abstract

Research motivation is missing.

Without detailed information about the methodology, it's challenging to assess the rigor of the study. If there are flaws or limitations in the agent-based model or the data used, it could undermine the credibility of the results.

The abstract lacks specific details about the model's structure, parameters, and assumptions. It's essential for a scientific paper to provide clear and transparent information about these aspects so that other researchers can assess and replicate the study.

Introduction

The COVID-19 pandemic has been a rapidly evolving crisis, and research conducted during the pandemic may quickly become outdated as the situation changes. As we transition into the post-pandemic phase, the relevance of research on the immediate impact and response strategies to COVID-19 diminishes.

Covid-19 is a transmitted virus which cannot be effectively forecasted by Agent-based models (ABM).

Therefore, the contributions claimed by the authors in page 3, row 144 onwards are not valid.

Results

The results that are forecasted by the various models are meaningless as they cannot reflect the true situation of Covid-19.

Conclusions

As previously mentioned, the paper's relevance may be diminished in a post-pandemic context, and the conclusion should address why the research findings remain significant and how they can contribute to ongoing public health discussions.

Comments on the Quality of English Language

adequate

Reviewer 2 Report

Comments and Suggestions for Authors

The literature review is not comprehensive. There is a lack of utilization of agent-based models for the assessment and simulation of the Covid-19 pandemic processes. Both the literature studies and comparative analyses with other authors require supplementation.

A list of exemplary publications in this field (selection is based on high citation rates and open access):

- Covasim: An agent-based model of COVID-19 dynamics and interventions, DOI: 10.1371/journal.pcbi.1009149

- Agent-Based Social Simulation of the Covid-19 Pandemic: A Systematic Review Download PDF, DOI: 10.18564/jasss.4601 

- High-Resolution Agent-Based Modeling of COVID-19 Spreading in a Small Town, DOI: 10.1002/adts.202000277

The presentation of results is comprehensive and does not require further additions. 

Reviewer 3 Report

Comments and Suggestions for Authors

The study of this manuscript is of great research significance, some contents are not detailed enough, the innovation is a little insufficient, the overall conclusions and suggestions are appropriate, and no specific measures are given, which has limitations.

Line 157, based on the study of the influence of COVID-19 on epidemiology, can specifically count the number of hospitals in Karaganda, not just "several".

The social network of Line 169-170 Figure1 assumes that different groups of people go to few and fixed places, which is ideal and cannot reflect the diversity of human activities in real life.

The time span of data collected by Line 244-247 is about 250 days, so the time span can be appropriately extended.

Line 413-421 This paragraph specifically lists the influence of the modeling results of BS, SC, MV and VS on the age, and it can be summarized at the end of the paragraph that "this does not mean that the elderly will necessarily play a leading role in the spread of COVID-19".

The subtitle of Line 422-435 is Economic impact of COVID-19, and the title is not refined enough. The follow-up content is to specifically deal with the cost of COVID-19 medical resources, and does not involve the impact of COVID-19 on other aspects of the economy, which is limited to the medical field.

Line 455-461 This paragraph mentions that "for this reason, infectious diseases departments in all hospitals have been expanded, and 459 wards have been newly opened", but it does not specifically describe the specific types of additional wards (general wards, intensive care wards, artificial ventilation wards), which can be specifically counted to simulate the needs of critically ill patients and critically ill patients for beds in different types of wards.

The conclusion of Line 505-522 puts forward the conclusion that "the simulation of various scenarios shows that the most important places of transmission of infection are families and public transport", which is not innovative and lacks novelty.

Line 529 points out that the measured data of children, young people and elderly patients are not consistent with the model results. The explanation is limited only from the susceptibility of children, and the reasons for the inconsistency can be explained from the susceptibility and morbidity of the elderly and the incidence of children.

The specific research conclusions of Line 584 can be subdivided into subheadings or marked with serial numbers, which will be more logical.

Comments on the Quality of English Language

The study of this manuscript is of great research significance, some contents are not detailed enough, the innovation is a little insufficient, the overall conclusions and suggestions are appropriate, and no specific measures are given, which has limitations.

Line 157, based on the study of the influence of COVID-19 on epidemiology, can specifically count the number of hospitals in Karaganda, not just "several".

The social network of Line 169-170 Figure1 assumes that different groups of people go to few and fixed places, which is ideal and cannot reflect the diversity of human activities in real life.

The time span of data collected by Line 244-247 is about 250 days, so the time span can be appropriately extended.

Line 413-421 This paragraph specifically lists the influence of the modeling results of BS, SC, MV and VS on the age, and it can be summarized at the end of the paragraph that "this does not mean that the elderly will necessarily play a leading role in the spread of COVID-19".

The subtitle of Line 422-435 is Economic impact of COVID-19, and the title is not refined enough. The follow-up content is to specifically deal with the cost of COVID-19 medical resources, and does not involve the impact of COVID-19 on other aspects of the economy, which is limited to the medical field.

Line 455-461 This paragraph mentions that "for this reason, infectious diseases departments in all hospitals have been expanded, and 459 wards have been newly opened", but it does not specifically describe the specific types of additional wards (general wards, intensive care wards, artificial ventilation wards), which can be specifically counted to simulate the needs of critically ill patients and critically ill patients for beds in different types of wards.

The conclusion of Line 505-522 puts forward the conclusion that "the simulation of various scenarios shows that the most important places of transmission of infection are families and public transport", which is not innovative and lacks novelty.

Line 529 points out that the measured data of children, young people and elderly patients are not consistent with the model results. The explanation is limited only from the susceptibility of children, and the reasons for the inconsistency can be explained from the susceptibility and morbidity of the elderly and the incidence of children.

The specific research conclusions of Line 584 can be subdivided into subheadings or marked with serial numbers, which will be more logical.

Round 2

Reviewer 1 Report

Comments and Suggestions for Authors

Review report healthcare-2609609 R1

THE EPIDEMIOLOGICAL AND ECONOMIC IMPACT OF COVID‑19 IN KAZAKHSTAN.

AGENT BASED MODELLING

Abstract

Research motivation is missing.

The authors claimed that “Unfortunately, the size of the abstract is limited by the journal's conditions, and we were not able to describe the methodology used in detail.” This reason is not accepted.

Writing an effective journal abstract within word limitations requires careful planning and concise communication of your research. Please ensure your abstract is comprehensible to someone who may not be familiar with your field. You should also emphasize what makes your study unique or innovative. Please mention any practical applications or future research directions.

What is the implication of this study?

Introduction

Covid-19 is a transmitted virus which cannot be effectively forecasted by Agent-based models.

Page 3, row 144. “We created a stochastic agent-based model of infection transmission for a large city in Kazakhstan. Some Kazakh scientists have already published COVID-19 models in the literature; however, as time has shown, the forecast data significantly differed from reality.” This sentence is hanging. So, what is next?

You have summarised so much pf past literature in the page 2 and 3, so what is the research gap?

Results

The forecasting model presented in this article does not significantly advance the state of knowledge or methodology.

Conclusions

Row 602. Since the authors claimed that “The measures taken by the state authorities of Kazakhstan were primarily aimed at saving people's lives. In our opinion, they were quite effective, since the incidence of COVID-19 in the republic was 8000/100,000 population and the mortality rate was 101/100,000 population, which is much lower than in many countries. This made it possible to provide health organizations with specialists and material resources in a timely manner.” It means that the government was doing adequately during the Covid-19. So, what is the reason to forecast Covid-19 in Kazakhstan?

Research that demonstrates real-world impact or policy relevance is more likely to be accepted and cited. Forecasting models that have practical applications or can inform public health decision-making may have an advantage. However, the practical applications from this article are missing.

Reviewer 3 Report

Comments and Suggestions for Authors

I suggest changing the modified section to red font, as the current revised version is difficult to read clearly.